# Genome-Wide Identification and Analysis of the MAPK and MAPKK Gene Families in Potato (*Solanum tuberosum* L.)

Yutong Shang [1,2,†], Xiaobo Luo [1,2,†], Heng Zhang [2], Mingjun Chen [2], Wang Yin [2], Zhenju Cao [2], Renju Deng [2], Yan Li [1,*] and Fei Li [2,*]

1   Key Laboratory of Plant Resource Conservation and Germplasm Innovation in Mountainous Region (Ministry of Education), Collaborative Innovation Center for Mountain Ecology & Agro-Bioengineering (CICMEAB), College of Life Sciences, Institute of Agro-Bioengineering, Guizhou University, Guiyang 550025, China
2   Guizhou Institute of Biotechnology, Guizhou Provincial Academy of Agricultural Sciences, Guiyang 550003, China
*   Correspondence: yli@gzu.edu.cn (Y.L.); lifeipotato@163.com (F.L.)
†   These authors contributed equally to this work.

**Abstract:** Mitogen-activated protein kinase (MAPK) is an important component of the signal transduction pathway, which plays important roles in regulating plant growth and development, and abiotic stress. Potato (*Solanum tuberosum* L.) is one of the most popular tuber crops in the world. Genome-wide identification and analysis of the *MAPK* and *MAPKK* gene family in potato is not clear. A total of 20 *MAPK* genes and 8 *MAPKK* genes were identified in the potato genome. A conservative motif analysis showed that the MAPK protein contained a typical TxY phosphorylation site, and the MAPKK protein contained a conservative characteristic motif S/T-x5-S/T. Phylogenetic analysis showed that potato *MAPK* (mitogen-activated protein kinase) and *MAPKK* (mitogen-activated protein kinase kinase) were similar to Arabidopsis, including four groups of members A, B, C and D. Gene structure and promoter sequence analysis showed that all 28 gene family members of potato *Solanum tuberosum MAPK* (*StMAPK*) and *StMAPKK* have coding regions (CDS), and family members in the same group have similar intron and exon compositions, and that most cis-acting elements upstream of gene promoters elements have related to stress response. Chromosome location analysis found that MAPKs were unevenly distributed on 11 chromosomes, while MAPKKs were only distributed on chromosomes Chr. 03 and Chr. 12. Collinearity analysis showed that *StMAPKK3* and *StMAPKK6* have the same common ancestors among potato, pepper, and tomato. qRT-PCR results showed that the relative expressions of *StMAPK14* and *StMAPKK2* were significantly upregulated under low-temperature stress. These results could provide new insights into the characteristics and evolution of the *StMAPK* and *StMAPKK* gene family and facilitate further exploration of the molecular mechanism responsible for potato abiotic stress responses.

**Keywords:** potato; MAPK; MAPKK; gene family; low-temperature stress

## 1. Introduction

Low-temperature stress limits the growth, development, and distribution of plants, which are divided into cold and freezing damage by the degree of zero. Cold damage refers to the damage caused by low temperatures above the degree of zero, which mainly affects the process of photosynthesis and respiratory metabolism of plants, resulting in the disorder of plant cell function. Frozen damage refers to the damage caused by low temperatures below the degree of zero, and freezing damage will cause plant cells to freeze and damage the plant cell membrane, causing plant cell damage or death [1,2]. Plant have evolved complex and elaborate mechanisms against temperature extremes during their long-term response to cold stress. Previous studies have shown that the signaling pathway with CBF (C-repeat-binding factors) transcription factor plays important roles in plants' fight against cold stress [2]. In Arabidopsis, the MAPK cascade pathway has

been reported to regulate the antifreeze capacity by working together with the upstream regulators of CBF [3].

The MAPK cascade pathway is a common and evolutionarily conserved signal transduction mode in all eukaryotes, which is mainly composed of MAPKKK, MAPKK and MAPK kinases [4,5]. When cells perceive signals through the proprioceptor or receptor and stimulate MAPKKK activation, they further regulate downstream reactions through substrates such as phosphorylated downstream transcription factors and protein kinases, and realize signal transduction [6,7]. It has been reported that the MAPK cascade plays key roles in regulating plant growth and development and in the response of plants to various stresses in the environment [8,9]. The function of the *MAPK* gene has been identified in many plants, such as in Arabidopsis, maize, cotton, and tobacco [10–13]. The involvement of YODA and mitogen-activated protein kinase 6 has been identified in Arabidopsis post-embryogenic root development through auxin upregulation and cell division plane orientation [14]. In Arabidopsis, brassinolide regulates stomatal development through GSK3-mediated inhibition of the MAPK pathway [11]. In maize leaves, ABA induces the production of hydrogen peroxide and activates MAPK by inducing the expression of antioxidant enzymes, forming a positive feedback signaling pathway [10]. MAPK regulates plant cellular immunity by phosphorylating the WRKY transcription factor in tobacco [12]. The overexpression of *GhMKK3* promoted root growth and ABA-induced stomatal closure, which in turn was involved in the response of cotton plants to drought stress [13]. It was found that flagellin *flg22* activated *MEKK1-MKK4/5-MPK3/6-WRKY22/29* by interacting with the plasma membrane receptor *Fls2*, which could endow plants with resistance to bacteria and fungi by promoting the expression of disease resistance genes in Arabidopsis [15]. *mpk3*, *mpk6* single mutants and *mpk3mpk6* double mutants were obtained by CRISPR technology and showed stronger cold resistance, while the expression of *mpk3/mpk6* weakened the cold resistance of plants. It has been proven that *MPK3* and *MPK6* interact with ICE1, and negatively regulates the expression of *CBF* gene in Arabidopsis and the frost resistance of Arabidopsis plants [3,16]. These results indicate that *MPK* play important roles in regulating the plant response to cold stress.

Potato (*Solanum tuberosum* L.) is the fourth largest grain crop and plays an important role in food security and economy in China and other countries in the world. Potato prefers coolness but is not resistant to low temperatures. In recent years, frequent freezing in extreme weather events has become one of the main limiting factors for potato production. A number of cold genes have been identified and characterized in potato, such as *SAD* [17], *ADC1* [18] and *SaCBL1-like* [19]. Overexpression of the *S. commersonii SAD* gene in cultivated potato variety Zhongshu 8 significantly increased freeze tolerance in cv. Zhongshu 8 [17]. A previous study indicated that the upregulation of *ADC1* expression increased the content of putrescine and enhanced the freezing tolerance of potato [18]. *Solanum acaule CBL1-like* contributed to increased freezing tolerance and the expression of *CBF1* in potato [19]. A total of 15 *StMAPKs* and five *StMAPKKs* were identified in potato, and overexpression of *StMKK2* led to a significant enhancement in the expression of *CBF1/2/3*, *OPR2* and *SLD2* and the cold tolerance of potato; *StMKK2* interacted with *StMAPK4/7* in the cytoplasm by yeast in two hybrid and two bimolecular fluorescence complementation experiments [9]. These results showed that *MAPKs* gene family played an important role in the response to cold stress in potato. However, the molecular mechanism of cold stress response in potato remains unclear.

In this study, MAPKs and MAPKKs were systematically identified, and phylogeny relationship, conserved motifs, gene structure, promoter analysis, chromosome location, collinearity analysis, protein characteristics were analyzed based on reference genome sequences of potato. Quantitative real-time PCR (qRT-PCR) was conducted to validate the gene expression of MAPKs and MAPKKs under low-temperature stress. The results will provide an important theoretical basis for the expansion and evolutionary history of the MAPKs and MAPKKs gene family, dissecting the molecular mechanism of potato low-temperature stress response.

## 2. Materials and Methods

### 2.1. Identification of MAPK and MAPKK Genes in Potato

The conserved domain of MAPK proteins (http://pfam.xfam.org/family/PF00069; accessed on 13 October 2021) was used as the probe to search StMAPK and StMAPKK sequnences in the potato genome database (http://spuddb.uga.edu/; accessed on 5 April 2021) using the Hidden Markov model search software (HMM) with the E-value set to 0.01 [20,21]. In order to verify the reliability of the predicted sequences, the candidate potato MAPK and MAPKK family sequences' information were submitted to NCBI (https://www.ncbi.nlm.nih.gov/; accessed on 25 November 2021) and Interproscan database (http://www.ebi.ac.uk/interpro/; accessed on 25 November 2021). The MAPK and MAPKK gene sequences of Arabidopsis were analyzed by TAIR database (http://www.arabidopsis.org/; accessed on 15 March 2022).

### 2.2. Proteins Sequence and Phylogenetic Analysis of StMAPK and StMAPKK

The multiple alignment of potato StMAPK and StMAPKK protein sequences was performed by an online website cluster (https://pir.georgetown.edu/pirwww/search/multialn.shtml; accessed on 3 December 2021), and the parameter value was set as the default. Unrooted phylogenetic trees of the 20 predicted StMAPK proteins and those of the reported 20 AtMAPK proteins and 8 StMAPKK proteins and 10 AtMAPKK proteins were generated using the MEGA 7 software, with 1000 bootstrap replicates (Test of phylogeny), p-distance (Model/Method), and pairwise deletion (Gaps/Missing Data treatment) [22].

### 2.3. Sequence Analysis of StMAPK and StMAPKK Genes

The intron and exon structure of the StMAPK and StMAPKK gene were obtained by comparing the coding sequence with the corresponding genomic DNA sequence by using Gene Structure Display Server (http://gsds.cbi.pku.edu.cn/; accessed on 3 December 2021) [23]. The MEME software (http://meme.sdsc.edu/meme/cgi-bin/meme.cgi; accessed on 3 December 2021) was used to analyze StMAPK and StMAPKK genes conserved motifs, where the maximum motif number was set to 10, the optimal the motif width was set to 6–50 [24]. In addition, PlantCARE (http://bioinformatics.psb.ugent.be/webtools/plantcare/html/; accessed on 6 December 2021) database was used to predict cis-acting elements of MAPK and MAPKK genes promoter region in potato.

### 2.4. Synteny Analysis and Chromosomal Localization

Collinearity analysis was performed using the method described in the PlantDGD (http://pdgd.njau.edu.cn:8080/; accessed on 20 March 2022) [25]. The Multiple Collinearity Scan toolkit (MCScanX) was used to identify the StMAPK and StMAPKK duplication events. BLASTP was performed to identify the intra-species paralogous pairs using protein sequences with the following parameters settings: (1) alignment significance: E_VALUE (default: $1 \times 10^{-5}$); (2) MATCH_SCORE: final score (default: 50) [26], and then the data were integrated and plotted using Circos [27]. According to the relative positions of the identified *MAPK* and *MAPKK* genes annotated in the potato genome, MapInspect software was used to map these genes on the potato chromosome.

### 2.5. StMAPKs and StMAPKKs Protein Characterization

The ProtParam program on the online software ExPASy website (http://web.expasy.org/protparam/; accessed on 15 December 2021) was used to predict StMAPK and StMAPKK protein signatures, including the number of amino acids (AA), theoretical molecular weight (MW), instability index (II), and isoelectric point (pI).

### 2.6. Expression Analysis of StMAPK and StMAPKK Family Members

To understand the response of StMAPK and StMAPKK family members to low-temperature stress, the low-temperature sensitive variety 'Desiree' was subjected to −2 °C (0 h, 1 h, 2 h, 4 h, 8 h, 16 h, 24 h, 48 h), the leaves of all treatments were collected. Total RNA

was isolated from the control and treated potato leaves using TaKaRa MiNiBEST Universal RNA Extraction kit (TaKaRa, Beijing, China). Then, the RNA was reverse transcribed into cDNA using StarScript II RT Mix with gDNA Remover (GeneStar, Beijing, China) according to the manufacturer's instructions. qRT-PCR was performed on a CFX96 touch real-time PCR detection system (Bio-Rad, Hercules, CA, USA) using ABM BlasTaq$^{TM}$2X qPCR MasterMix. The qRT-PCR system was 20 μL, including 0.5 μL of forward primer and reverse primer, 1 μL of cDNA, 10 μL of BlasTaqTM2X qPCR MasterMix, and 8 μL of ddH$_2$O. The concentration of cDNA was 50 ng μL$^{-1}$, and the concentration of primers was 10 μL μmol L$^{-1}$. The reaction conditions were: pre-denaturation at 95 °C for 3 min, denaturation at 95 °C for 15 s, annealing at 60 °C for 1 min, and the reaction was 40 cycles. *βActin* was used as the internal reference gene, the Ct ($2^{-\Delta\Delta Ct}$) method was used to calculate the relative expression of the gene. The experiment was performed with three technical replicates and three biological replicates. Specific primers for *StMAPK* and *StMAPKK* genes were designed using Beacon Designer 7.7 (Premier Biosoft International, Palo Alto, CA, USA) (Table S1).

## 3. Results

### 3.1. Identification of MAPK and MAPKK Genes and Multiple Alignment Analysis in Potato

A total of 28 candidate MAPK and MAPKK family members, including 8 StMAPKKs and 20 StMAPKs were obtained in potato genomes using HMM software. The 28 potato MAPK and MAPKK genes were named StMAPK1 to StMAPK20, StMAPKK1 to StMAPKK8 according to the comparison results of NCBI and the homology relationship with the annotated MAPK and MAPKK family members of Arabidopsis (Table S2). In order to further evaluate the phylogenetic relationship between MAPK and MAPKK genes in potato, the full-length protein sequences of 20 StMAPKs and 8 StMAPKKs were analyzed by multiple comparisons using the online website cluster W [28]. Multiple alignment analysis showed that the conserved catalytic domain T(E/D)YVxTRWYRAPE(L/V) were detected in the potato StMAPKs protein sequence, in which the members of StMAPKs in groups A, B, and C contained TEY activation domains, while StMAPKs belonging to members of group D all possessed TDY activation domains (T: threonine, E: glutamic acid, Y: tyrosine, D: aspartic acid). The D group members in potato were also existed Arabidopsis and alfalfa. The unique catalytic domain VGTxxYM(S/A) PEG in the sequence of StMAPKKs protein was identified, and another conservative characteristic motif S/T-x5-S/T, as a phosphate site, also existed in wheat, Arabidopsis, and rice [29]. It could be shown that the evolutionary conservation of MAPKKs provides a basis for further functional verification (Figure 1).

### 3.2. Phylogenetic Analysis of the MAPK and MAPKK Proteins

To better understand the phylogenetic relationship between StMAPK and StMAPKK, the 20 reported Arabidopsis MAPK proteins, 10 MAPKK proteins, and the identified amino acid sequences of StMAPK and StMAPKK family members were selected to construct an unrooted phylogenetic tree by NJ method. As shown in 40 MAPK proteins from Arabidopsis and potato were divided into four groups (Group A–D). The StMAPK14 and StMAPK20 proteins belong to group A, StMAPK4, StMAPK15 and StMAPK18 belong to group B, group C contains StMAPK5, StMAPK6 and StMAPK7, and the remaining 12 StMAPK proteins all belong to group D. The MAPKK protein is also divided into four groups, where groups C and D have one member each, group A contains StMAPKK2 and StMAPKK3, and group B has members of StMAPKK4, StMAPKK5, StMAPKK6, and StMAPKK8, which constitute half of the whole potato StMAPKK protein (Figure 2).

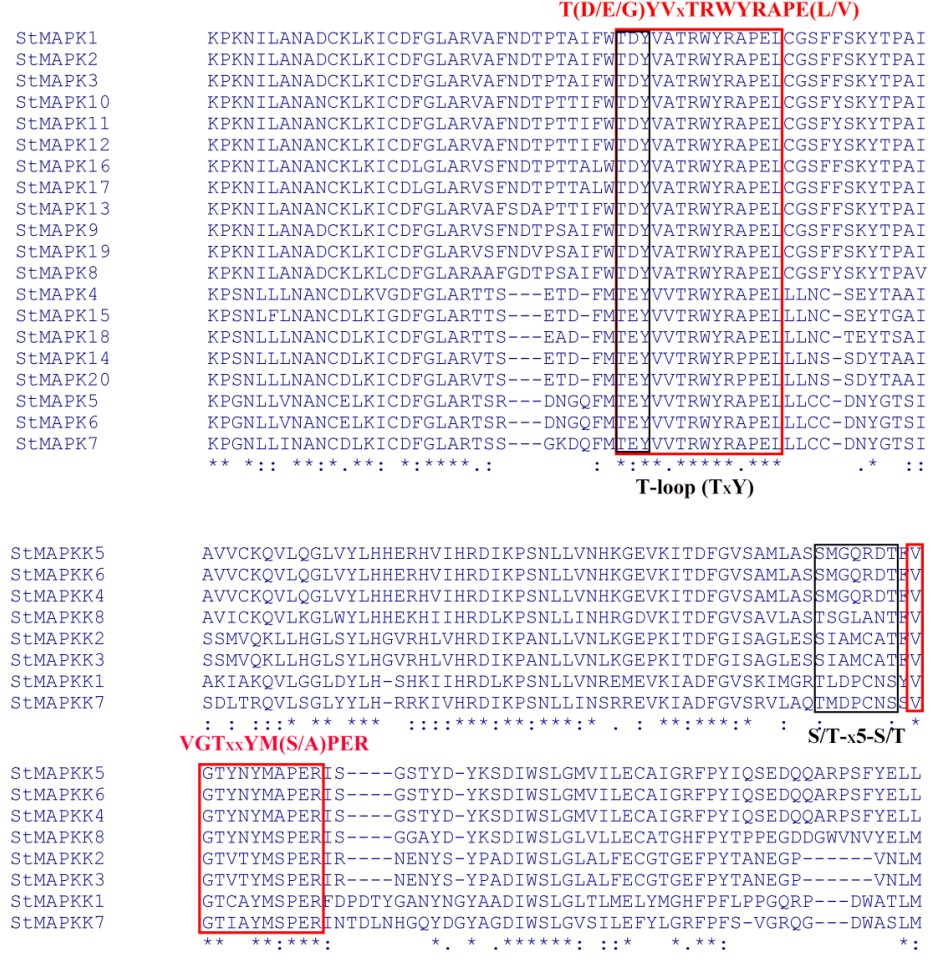

**Figure 1.** Multiple alignment of amino acid sequences between MAPK and MAPKK family members in potato.

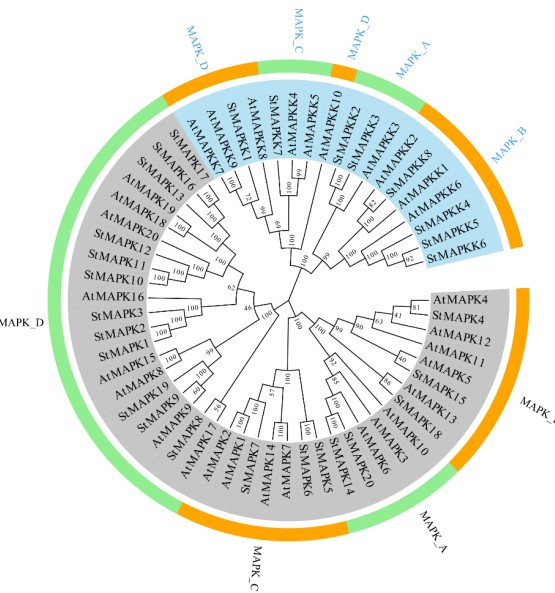

**Figure 2.** Phylogenetic relationships between Arabidopsis and potato MAPK and MAPKK family members.

### 3.3. MAPK and MAPKK Gene Structure and Conserved Motif Analysis

To better understand the structure of the StMAPK and StMAPKK genes, the CDS sequences of the StMAPK and StMAPKK genes were aligned to the genomic sequence. According to the gene structure analysis, all 28 gene family members of potato StMAPK and StMAPKK have coding regions (CDS), and StMAPK8 and StMAPK20 do not contain untranslated regions (UTR), the intron sequence length of StMAPK14 is long. At the same time, StMAPK and StMAPKK with higher sequence similarity in structure, such as belonging to StMAPKK B group members, except for StMAPKK8, the coding regions of other genes are similar in length and structure. Therefore, compared with other subfamilies, the gene structure in StMAPKK B group is more conserved. In addition, the results showed that the structures of StMAPK and StMAPKK genes in the same evolutionary branch were roughly similar, and the numbers of exons and introns are basically the same, but the lengths of exons and introns are different (Figure 2).

In order to further analyze the conserved domains of MAPK and MAPKK proteins, the online software MEME was used to perform motif analysis on 20 Arabidopsis MAPK proteins, 10 MAPKK proteins. A total of 10 motifs were predicted in the StMAPK and StMAPKK proteins. All members of the StMAPK protein belonging to group D contained all 10 conserved motifs and had the same order. Except for StMAPK14 in group A, the remaining three groups did not contain motif 8 and motif 10. Among the StMAPKK proteins, group A only contained five members, motif 4, motif 1, motif 5, motif 2 and motif 6, while StMAPKK8 belonging to group B contained two conserved motifs, and StMAPKK5 belonging to group B did not have three groups B, C and D. Most of the MAPK and MAPKK proteins in the same evolutionary clade had similar motif distributions (Figure 3). In addition, the motif patterns on each evolutionary branch were both conservative and specific, suggesting that MAPK and MAPKK proteins from the same evolutionary branch may have similar functions, while different motif patterns between different evolutionary branches prove the functional differences on different branches.

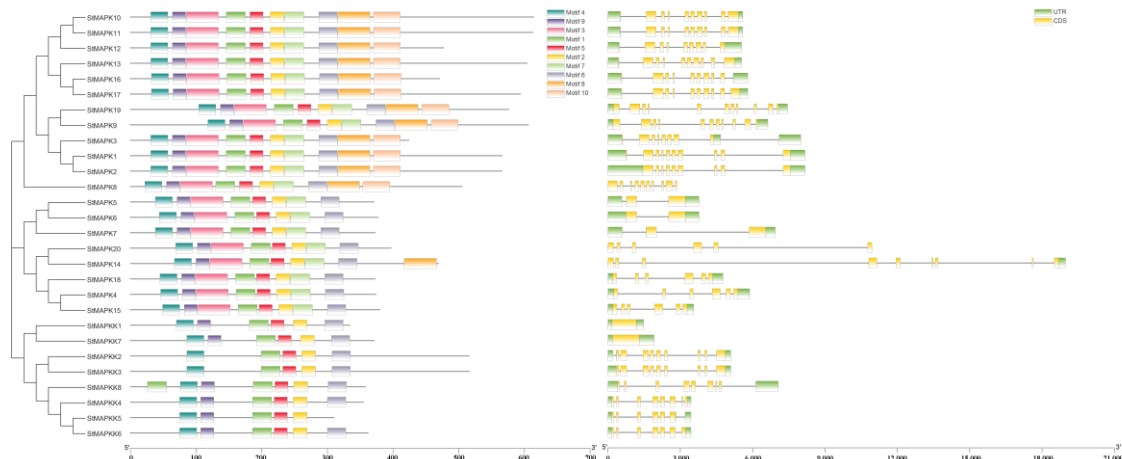

**Figure 3.** Analysis of gene structure and conserved motifs of potato MAPK and MAPKK gene family.

### 3.4. Chromosome Location of StMAPK and StMAPKK Genes

The 28 candidate StMAPK and StMAPKK genes predicted in this study were all mapped to the potato chromosome (Figure 4). The results showed that 28 genes were widely distributed on 11 chromosomes from Chr. 01 to Chr. 12 without Chr. 09, among them, StMAPKs are scattered on all chromosomes except Chr. 03, while the 7 StMAPKKs of StMAPKK are all on the Chr. 03 chromosome, and the Chr. 12 chromosome contains both StMAPK and StMAPKK. In general, StMAPKs were mainly distributed at the top or end of chromosomes, 4 StMAPKs genes located on Chr. 07 were distributed at the end and 2 StMAPKs genes located on Chr. 10 were distributed at the top. However, all genes of StMAPKKs were distributed at the end of chromosomes. The duplication events play an

crucial role in promoting the expansion of plant gene families. In this study, a total of three pairs of StMAPK and StMAPKK genes were collinear by MCScanX analysis, indicating genome duplication events event might play an essential role in the expansion of StMAPK and StMAPKK gene family (Figure 5).

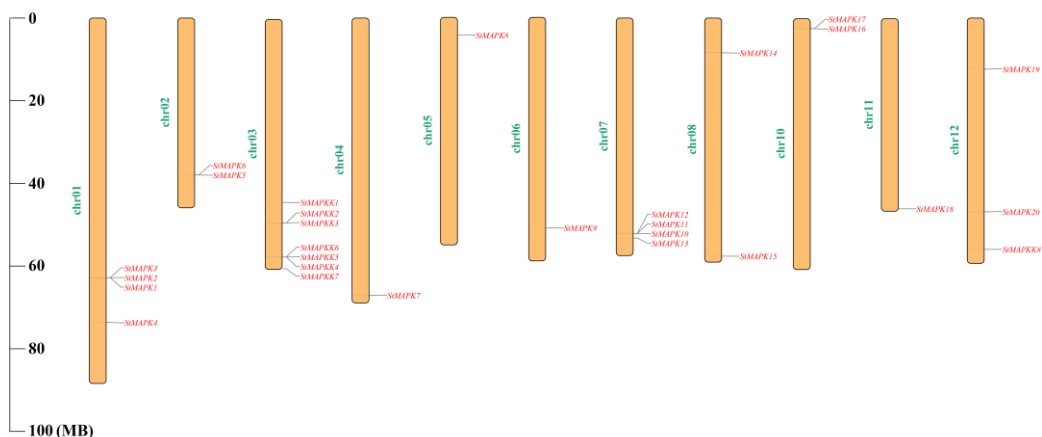

**Figure 4.** Chromosome location of StMAPK and StMAPKK genes.

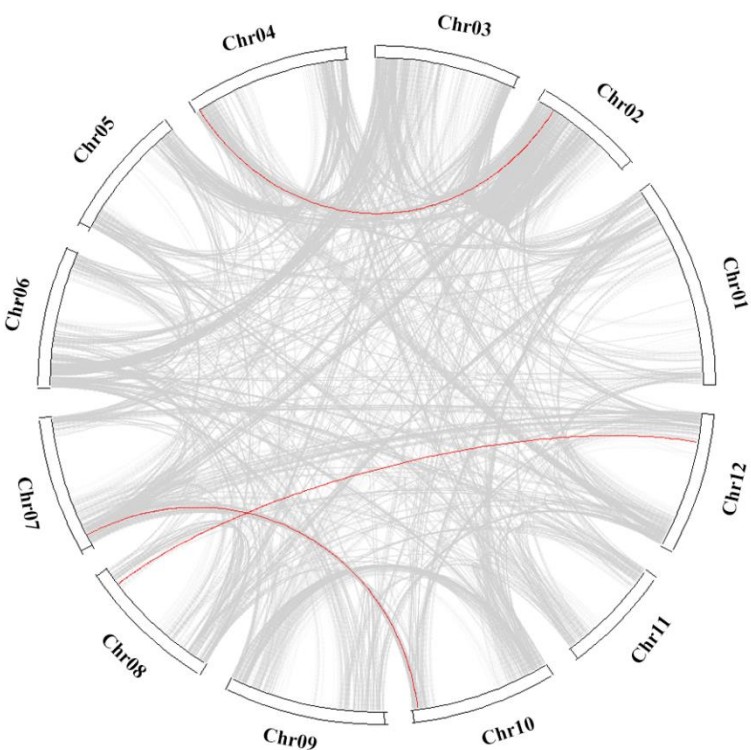

**Figure 5.** Collinearity analysis of the StMAPK and StMAPKK family.

### 3.5. Promoter Sequence Analysis

Many studies have shown that some cis-acting elements in the gene promoter region play key roles in the process of gene transcription regulation. In this study, the promoter sequences of all StMAPK and StMAPKK genes (within 2 kb upstream of the start codon) were analyzed. The results showed that the promoter region of StMAPK and StMAPKK genes contained many basic elements involved in important biological processes, such as ACE, BOX I and GATA motif in photoresponse (Table S3). The TATC box, TGA element, TGACG motif and other elements related to plant hormone response also existed in the promoter region of StMAPK and StMAPKK genes. In addition, a large number of cis-acting elements related to plant stress were present in the promoter region of StMAPK and

StMAPKK genes, such as ABRE related to ABA response, MBS related to drought stress response, HSE related to high temperature stress, LTR related to low-temperature stress, and TC-rich repeats related to stress. These results suggest that StMAPK and StMAPKK genes may play important roles in light, hormones and abiotic stress.

### 3.6. Prediction of StMAPK and StMAPKK Protein Sequence Features

The physical and chemical properties of 20 StMAPK and 8 StMAPKK proteins were analyzed with ExPASy (Table S4). Among these, the StMAPK protein sizes varied from 370 to 613 AAs with molecular weights (MWs) from 42.68 to 69.71 KDa, respectively. Additionally, the theoretical isoelectric point (PI) of StMAPK protein ranged from 4.97 to 9.37, of which 12 were alkaline with PI values greater than 7, and the other eight are acidic. More than half of the StMAPKs were classified as unstable proteins according to the instability index. The prediction results of the fat index showed that most StMAPK proteins contained a large number of fatty amino acids. The prediction of hydrophilicity analysis showed that all StMAPK proteins are predicted to be hydrophilic proteins. However, the number of amino acids of StMAPKK protein was not different, and stable between 307 and 309 except for StMAPKK3 and StMAPKK6. Only StMAPKK3 and StMAPKK8 were alkaline. Also, more than half of the StMAPKKs were unstable proteins and all the StMAPKK proteins were hydrophilic. There was a conservation among the most StMAPKs and StMAPKKs members, while different AA sequences in non-conserved regions may alter some of the molecular characteristics.

### 3.7. Collinearity of StMAPK and StMAPKK with Solanaceae Species

To further analyze the origin and evolutionary history of MAPK and MAPKK family genes, a collinear relationship between *MAPK* and *MAPKK* genes of pepper and tomato was constructed. A total of 41 pairs of collinear genes were obtained, including 15 *StMAPK* and *StMAPKK* genes and 32 *CaMAPK* and *CaMAPKK* genes. Among them, three pairs of orthologous genes (one to one) were identified, including Capana03g001606 (*CaMKK9*) and *StMAPKK3*, Capana03g001444 (*CaMKK3*) and *StMAPKK6*, Capana00g004661 (*CaMAPK9-2*) and *StMAPK19*. A total of 39 pairs of collinear genes were obtained, including 16 *StMAPK* and *StMAPKK* genes and 35 *ScMAPK* and *ScMAPKK* genes. Three pairs of orthologous genes were detected, including Solyc03g019850 (*ScMAPKK5*) and *StMAPKK6*, Solyc03g097920 (*ScMAPKK4*) and *StMAPKK3*, Solyc01g080240 (*ScMAPK13*) and *StMAPK15*. The results showed that *StMAPKK3* and *StMAPKK6* have the same common ancestors among potato, pepper, and tomato. In addition, one *StMAPK* and *StMAPKK* gene corresponded to multiple pepper and tomato genes, and multiple *StMAPK* and *StMAPKK* genes corresponded to pepper and tomato genes. These synchronous events suggested that some *MAPK* and *MAPKK* genes were evolved before the divergence of the potato, pepper, and tomato lineages (Figure 6).

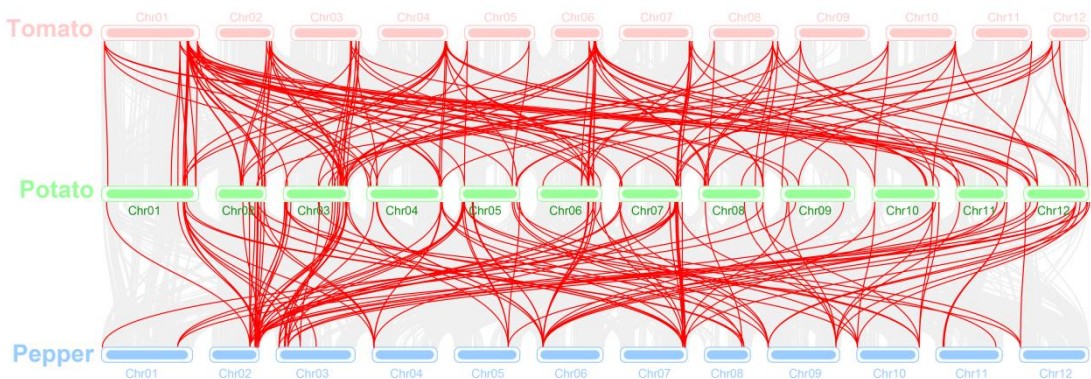

**Figure 6.** Synteny blocks of *MAPK* and *MAPKK* genes among potato, tomato and pepper.

### 3.8. Expression Analysis of StMAPK and StMAPKK under Low-Temperature Stress

The expression levels of StMAPKs and StMAPKKs genes were analyzed by qRT-PCR technology. The results showed that most of StMAPKs and StMAPKKs were significantly differential expressed under cold stress. StMAPK1, StMAPK5, StMAPK6, StMAPK9 were not significantly expressed. However, the relative expression level of StMAPK13 was downregulated. The expression levels of StMAPK4, StMAPK7, and StMAPK10 were significantly increased at 4 h of low-temperature treatment, and the expression of StMAPK8 was increased at 2 h. StMAPK14 was expressed at 8 h of low-temperature treatment. The relative expression levels of StMAPKK1, StMAPKK2, StMAPKK7, and StMAPKK8 were all upregulated. The above results are consistent with the previous conclusions that the members of group A show high levels of expression in the process of plant emergency stress [30] (Figure 7).

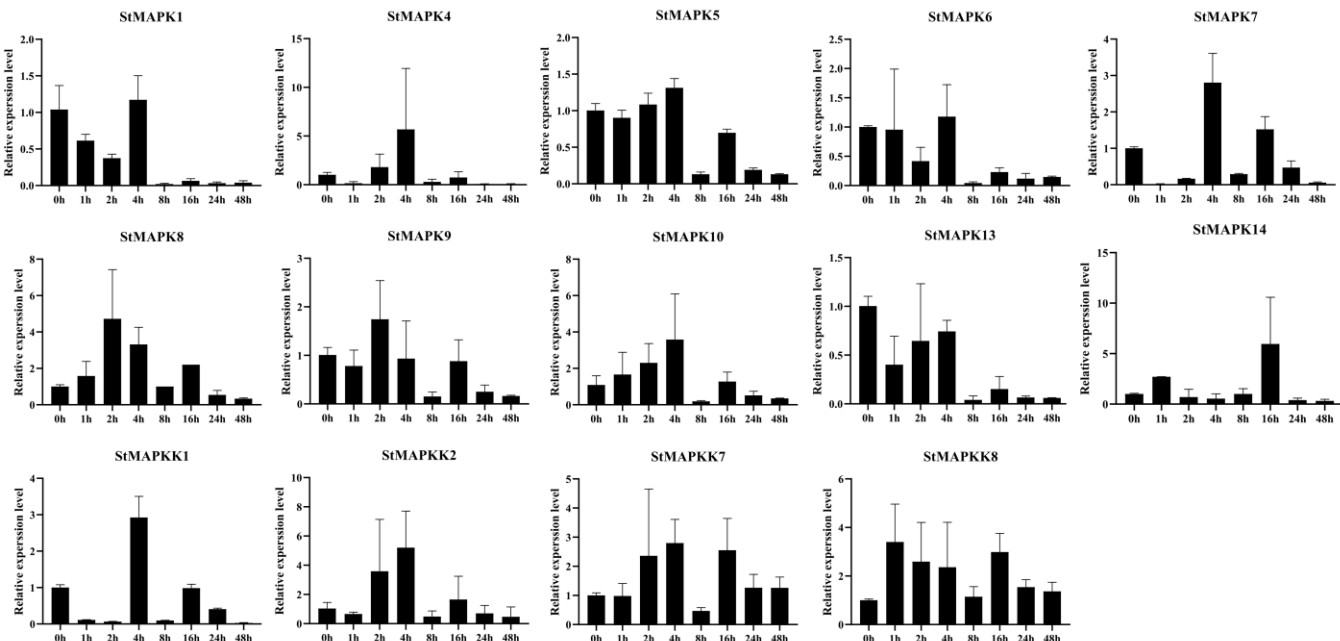

**Figure 7.** Expression pattern of *StMAPK* and *StMAPK*K after cold stress treatment at 2 °C.

## 4. Discussion

### 4.1. Characterization and Evolution of StMAPKs and StMAPKKs in Potato

Plant MAPK and MAPKK gene families play important roles in growth and development, hormone signal response, and biological and abiotic stress [5,31–35]. With the development of high-throughput sequencing technology and the improvement of plant genome sequencing, MAPK and MAPKK gene families have been identified in Arabidopsis, rice, poplar, melon, and wheat [29,36–38]. Although the MAPK family has been identified in potato [9], the characterization of the MAPK family is still not sufficient. In this study, 20 *StMAPK* and 8 *StMAPKK* genes were identified based on the latest potato genome. In this study, all canonical MAPKs were characterized by conserved TxY activation domains [39,40]. As previously reported, groups A, B and C contained the TEY activation domain, while group D contained the TDY activation domain (Figure 2). The presence of specific conserved motifs and structures in each group confirms the homologous evolution of MAPKs from a common ancestor. Previous studies have shown that MAPKs A members showed high levels of expression in plant growth and development and biological and abiotic stress [30]. MAPK3/6, which belongs to group A, performs important functions in the cold resistance of Arabidopsis [3]. The conserved characteristic motif S/T-x5-S/T of MAP-KKS, as a phosphate site, also exists in Arabidopsis and rice [30,41]. Arabidopsis MKK2 belonging to group B is involved in disease resistance and salt stress [42,43]. Based on the

functions of known protein kinases reported in plants, the role of homologous StMAPKs and StMAPKKs in potato response to abiotic stress and disease resistance is inferred. Protein structure analysis will be of great help to future functional research. The stage numbers of exons and introns in the same group are similar, and MAPKs in all reported species in group B show that they contain nearly six exons, while members of group C have less than 4 exons. The results showed that all three members of group B contain six exons, and the three members of group C all contain two exons [44–46]. However, members belonging to groups C and D in MAPKKs do not contain introns. In this study, the members of groups C and D of potato StMAPKs had the same structure [47–49]. The distribution of the exon–intron structure showed that the gene structure between homologous genes is basically conservative, and the MAPK and MAPKK genes of most species have the same origin in evolution.

### 4.2. Expression Profile Analysis of Cold Resistance-Related StMAPK and StMAPKK Genes

Extensive evidence has demonstrated that the MAPK gene family plays key roles in plant chilling stress-signaling pathways [50,51]. The expression of OsMPK4 and OsMSRMK2, and OsMPK3 can be activated by moderate low temperatures (12 °C) in rice [50–52]. In Arabidopsis, MEKK1-MKK2 is located upstream of MPK4 and MPK6 and plays a key role in the activation of MPK4 and MPK6 activities at low temperatures. Previous studies suggested that the MKK2 mutant showed a freeze–thaw sensitive phenotype under low-temperature conditions, overexpression of MKK2 showed a freeze–thaw resistant phenotype, and the MEKK1-MKK2-MPK4/MPK6 cascade signaling system plays an important role in plant response to low temperatures [43]. MPK3, MPK4 and MPK6, were rapidly activated after cold treatment and the MKK5-MPK3/MPK6 cascade promoted ICE1 degradation through the phosphorylation of Ser94, Thr366 and Ser403, negatively regulating cold stress. The MEKK1-MKK2-MPK4 cascade positively regulates the cold response, and constitutively inhibits the protein levels and kinase activities of MPK3 and MPK6 in Arabidopsis [51]. In maize, ZmMPK3 participated in the response to cold stress, and its transcription level increased significantly within 30 min [53]. Ectopic expression of ZmMKK1 improves tolerance to low-temperature stress through the accumulation of higher antioxidant enzyme activity, more osmo-regulatory substances, and significantly upregulated the expression of ROS-related and stress-responsive genes [54,55]. The relative expression levels of SlMAPK1/2/3 and SlCBF1 treated with $H_2O_2$ were higher than those in the control, and enhanced the chilling tolerance in tomato by regulating phytohormone concentrations and antioxidant enzyme activities [56]. In this study, the expression of StMAPK13 was down regulated in all time samples, especially at 8 h. The expression of StMAPK4, StMAPK7, StMAPK8, StMAPK10, StMAPK14 was upregulated. The relative expression of StMAPKK1, StMAPKK2, StMAPKK7, and StMAPKK8 were upregulated. Previous studies indicated that StMKK2 enhanced cold tolerance in potato by increasing the expression of CBF1/2/3, OPR2 and SLD2 [9]. Therefore, MAPKs and MAPKKs with differentially expressed genes may play important roles in response to cold stress.

### 5. Conclusions

In this study, 20 *StMAPK* and 8 *StMAPKK* genes were identified in potato genome. Phylogenetic analysis showed that all the MAPK and MAPKK proteins were divided into four groups of *StMAPK* and *StMAPKK* with a higher sequence similarity in gene structure. All members of the StMAPK protein belonging to group D contained all 10 conserved motifs and had the same order, except for StMAPK14. Many motifs and elements located in the promoter of *StMAPK* and *StMAPKK* genes were associated with light, hormones, and abiotic stress. Six *MAPK* and four *MAPKK* genes were differentially expressed under cold stress by qRT-PCR analysis. There results could provide insight into *StMAPK* and *StMAPKK* genes involvement in cold stress and provide a theoretical basis for cultivating cold-resistant varieties of potato.

**Supplementary Materials:** The following supporting information can be downloaded at: https: //www.mdpi.com/article/10.3390/agronomy13010093/s1, Table S1: qRT-PCR primer sequences; Table S2: MAPK and MAPKK family gene in potato; Table S3: The cis-elements responsive to stresses and hormones in the promoter regions of StMAPK and StMAPKK genes; Table S4: Physical and Chemical Properties of StMAPK and StMAPKK protein.

**Author Contributions:** Conceived and designed the experiments: Y.L. and F.L.; performed the experiments and analyzed the data: Y.S. and X.L.; wrote the manuscript: Y.L. and Y.S.; revised the paper: H.Z., M.C., W.Y., Z.C. and R.D. All authors have read and agreed to the published version of the manuscript.

**Funding:** This work was supported by Guizhou science and technology innovation talent team ([2020]5002), the National Natural Science Foundation of China (Grant No. 31960445), Guizhou Province Science and Technology Foundation ([2020]1Z015), and National potato industry technology system Guiyang comprehensive experimental station (2021–2022).

**Conflicts of Interest:** The authors declare no conflict of interest.

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
