# Peer review of "Genome-Wide Identification and Analysis of the MAPK and MAPKK Gene Families in Potato (Solanum tuberosum L.)"

_agronomy, doi:10.3390/agronomy13010093_

Round 1
Reviewer 1 Report
In this study, 20 StMAPK and 8 StMAPKK genes were identified in potato genome. Phylogenetic analysis of all the MAPK and MAPKK proteins were presented. And then, Six MAPK and four MAPKK genes were differentially expressed under cold stress by qRT-PCR analysis.
In my opinion the study does not go far enough.
1) Other experiment, such as Subcellular localization, Interaction network and Transgenic, should be carried out.
2) In addition, the CharacteristicsLength (Mw, kDa, pI , EF-hands, and so on) of MAPK and MAPKK genes should be described and discussed.
3) The Bootstrap value should be represented on the Phylogenetic tree.
4) All Methods description is not clear, more detail should be provided in “Materials and Methods”
So, the manuscript must go through a major overhaul. The author should make great efforts to improve the quality of the article.
Author Response
Point-Wise Responses
Manuscript ID: agronomy-2063302
Title: Genome-wide identification and analysis of the MAPK and MAPKK gene families in potato (Solanum tuberosum L.)
Response: Many thanks for the reviewer’s constructive suggestions. We have carefully checked the whole manuscript and made amendments in the revised version of the manuscript according to the reviewer’s suggestions. Specifically, some revisions are listed as follows:
- Other experiment, such as Subcellular localization, Interaction network and Transgenic, should be carried out.
Response: Many thanks for the reviewer’s constructive suggestion. Although the experiment of subcellular localization, interaction network and transgenic was not carried out in this study, some StMAPKs and StMAPKKs gene were significantly differential expressed under cold stress, such as StMAPKK1, StMAPKK2, StMAPKK7, and StMAPKK8 were all up-regulated. Previous studies indicated that overexpression of StMKK2 leads to a significant enhanced in the expression of CBF1/2/3, OPR2 and SLD2 and the cold tolerance of potato, StMKK2 interacts with StMAPK4/7 in the cytoplasm by yeast two hybrid and two bimolecular fluorescence complementation experiments (Chen et al; 2022). These results showed that MAPKs gene family played an important role in the response to cold stress in potato. Currently, we have successful performed the overexpression vector of 35S::pBI121::StMAPKK1, genetic transformation of 35S::pBI121::StMAPKK1 is doing now. We hope to obtain the results of transgenic experiments after the gene family analysis. It is expected to elucidate the molecular mechanism of genes involved in potato low temperature stress.
Reference
Chen, Y.; Chen, L.; Sun, X.M.; Kou, S.; Liu, T.T.; Dong, J.K.; Tu, W.; Zhang, Y.L.; Song, B.T. The mitogen-activated protein kinase kinase MKK2 positively regulates constitutive cold resistance in the potato. Environ Exp Bot. 2022, 194, 104702. https://doi.org/10.1016/j.envexpbot.2021.104702
- In addition, the Characteristics Length (Mw, kDa, pI, EF-hands, and so on) of MAPK and MAPKK genes should be described and discussed.
Response: Thanks for the reviewer’s suggestion. We revisit a detailed description and discussion of the StMAPK and StMAPKK gene properties.
Accordingly, the Paragraph 3.6 was revised as “The physical and chemical properties of 20 StMAPK and 8 StMAPKK proteins were analyzed with ExPASy (Table S4). Among these, the StMAPK protein sizes varied from 370 to 613 AAs with molecular weights (MWs) from 42.68 to 69.71 KDa, respectively. Additionally, the theoretical isoelectric point (PI) of StMAPK protein ranged from 4.97 to 9.37, of which 12 are alkaline with PI values greater than 7, and the other 8 are acidic. More than half of StMAPKs were classified as unstable proteins according to the instability index. The prediction results of fat index showed that most StMAPK proteins contain a large number of fatty amino acids. The prediction of hydrophilicity analysis shows that all StMAPK proteins are predicted to be hydrophilic proteins. However, the number of amino acids of StMAPKK protein was not different, stable between 307 and 309 except StMAPKK3 and StMAPKK6. And only StMAPKK3 and StMAPKK8 are alkaline. Also more than half of the StMAPKKs are unstable proteins and all of the StMAPKK proteins are hydrophilic. There is a conservation among the most StMAPKs and StMAPKKs members, while different AA sequences in non-conserved regions may alter some of the molecular characteristics. (Now in lines 275-289).
- The Bootstrap value should be represented on the Phylogenetic tree.
Response: Thanks for the reviewer’s suggestion. The Phylogenetic tree (Figure 2) with bootstrap value was provided in lines 208.
- All Methods description is not clear, more detail should be provided in “Materials and Methods”
Response: Many thanks for the reviewer’s constructive suggestion. We re-edited the materials and methods to ensure that they were clearer.
Accordingly, the sentence in lines 106-109 “The potato genome sequences (http://spuddb.uga.edu/) was used to identify MAPK and MAPKK genes. Hidden Markov model search software (HMM) was used to identified the conserved domain of MAPK protein (http://pfam.xfam.org/family/PF00069), the E-value was set to 0.01 [20-21].” was revised as “The conserved domain of MAPK proteins (http://pfam.xfam.org/family/PF00069; accessed on 13 October 2021) was used as the probe to search StMAPK and StMAPKK sequnences in the potato genome database (http://spuddb.uga.edu/; accessed on 5 April 2021) using the Hidden Markov model search software (HMM) with the E-value set to 0.01 [20-21].” (Now in lines 108-112).
the line 112 “(https://www.ncbi.nlm.nih.gov/)” was revised as “(https://www.ncbi.nlm.nih.gov/; accessed on 25 November 2021)” (Now in line 114).
the line 113 “(http://www.ebi.ac.uk/interpro/)” was revised as “(http://www.ebi.ac.uk/interpro/; accessed on 25 November 2021)” (Now in line 115).
the line 114 “(http://www.arabidopsis.org/)” was revised as “(http://www.arabidopsis.org/; accessed on 15 March 2022)” (Now in line 117).
the line 118 “(https://pir.georgetown.edu/pirwww/search/multialn.shtml)” was revised as “(https://pir.georgetown.edu/pirwww/search/multialn.shtml; accessed on 3 December 2021)” (Now in lines 121-122).
the sentence in lines 118-121 “and the parameter value was set as the default. The evolutionary tree was constructed with an adjacency method (NJ) by Mega 7.0 software, the bootstrap value was set to 1000, other parameters were still set to the default value [22].” was revised as “and the parameter value was set as the default. Unrooted phylogenetic trees of the 20 predicted StMAPK proteins and those of the reported 20 AtMAPK proteins and 8 StMAPKK proteins and 10 AtMAPKK proteins were generated using the MEGA 7 software, with 1000 bootstrap replicates (Test of phylogeny), p-distance (Model/Method), and pairwise deletion (Gaps/Missing Data treatment) [22].” (Now in lines 122-126).
the Paragraph 2.3 was revised as “The intron and exon structure of the StMAPK and StMAPKK gene were obtained by comparing the coding sequence with the corresponding genomic DNA sequence by using Gene Structure Display Server (http://gsds.cbi.pku.edu.cn/; accessed on 3 December 2021) [23]. The MEME software (http://meme.sdsc.edu/meme/cgi-bin/meme.cgi; accessed on 3 December 2021) was used to analyze StMAPK and StMAPKK genes conserved motifs, where the maximum motif number was set to 10, the optimal the motif width was set to 6-50 [24]. In addition, PlantCARE (http://bioinformatics.psb.ugent.be/webtools/plantcare/html/; accessed on 6 December 2021) database was used to predict cis acting elements of MAPK and MAPKK genes promoter region in potato.” (Now in lines 128-137).
the line 132 “(http://pdgd.njau.edu.cn:8080/)” was revised as “(http://pdgd.njau.edu.cn:8080/; accessed on 20 March 2022)” (Now in line 140).
the sentence in lines 132-133 “The collinearity block in StMAPK and StMAPKK genes were identified based on duplication events using MCScanX [26].” was revised as “The Multiple Collinearity Scan toolkit (MCScanX) was used to identify the StMAPK and StMAPKK duplication events. BLASTP was performed to identify the intra-species paralogous pairs using protein sequences with the following parameters settings: 1) alignment significance: E_VALUE (default: 1e-05); 2) MATCH_SCORE: final score (default: 50) [26].” (Now in lines 140-144).
the Paragraph 2.5 was revised as “The ProtParam program on the online software ExPASy website (http://web.expasy.org/protparam/; accessed on 15 December 2021) was used to predict StMAPK and StMAPKK protein signatures, including the number of amino acids (AA), theoretical molecular weight (MW), instability index (II), and isoelectric point (pI).” (Now in lines 149-152).
the sentence in lines 146-148 “Total RNA of all samples was extracted by RNA extraction kit. First-strand cDNA synthesis was performed using the genstar kit.” was revised as “Total RNA was isolated from the control and treated potato leaves using TaKaRa MiNiBEST Universal RNA Extraction kit (TaKaRa, Beijing, China). Then, the RNA was reverse transcribed into cDNA using StarScript II RT Mix with gDNA Remover (GeneStar, Beijing, China) according to the manufacturer’s instructions.” (Now in lines 156-160).
the sentence in lines 156-157 “qRT-PCR primers were synthesized by Tsingke Biotechnology Co., Ltd (Table S1).” was revised as “Specific primers for StMAPK and StMAPKK genes were designed using Beacon Designer 7.7 (Premier Biosoft International, Palo Alto, CA, USA) (Table S1).” (Now in lines 169-171).

Reviewer 2 Report
I read the manuscript entitled “Genome-wide identification and analysis of the MAPK and MAPKK gene families in potato ". In this study, the authors studied different bioinformatic analyses such as Gene structure and promoter sequence analysis, motif analysis, Cis-regulatory elements, and expression analysis. I think the author has written a paper well and needs to make minor changes before accepting the manuscript in the Agronomy journal.
1. In the title, please write the botanical/technical of the potato
2. L 17, please write the abbreviation of MAPKK
3. L 23, please write the abbreviation of stMAPK
4. In the whole article, the genes names and botanical names of crops/plants must be italic, and protein names must be non-italic
5. L 97, Quantitative real-time PCR (qRT-PCR)
6. Material and Methods; the author must write the accessed date of online websites or software, e.g. 05 December 2022, when the author used the online website.
7. L 145 needs space between 16h, 24h, etc.; please check the whole MS.
8. L 156-157, no need to write about the company; just write about software, which software or how to design the primers for expression analysis.
9. L 164, please delete one dot
10. Please carefully read the MS and check the space, dot, etc.
Author Response
Point-Wise Responses
Manuscript ID: agronomy-2063302
Title: Genome-wide identification and analysis of the MAPK and MAPKK gene families in potato (Solanum tuberosum L.)
Response: Many thanks for the reviewer’s constructive suggestions. We have carefully checked the whole manuscript and made amendments in the revised version of the manuscript according to the reviewer’s suggestions. Specifically, some revisions are listed as follows:
- In the title, please write the botanical/technical of the potato
Response: Many thanks for the reviewer’s constructive suggestion. The title of this article was revised as “Genome-wide identification and analysis of the MAPK and MAPKK gene families in potato (Solanum tuberosum L.)”. (Now in lines 2-3).
- L 17, please write the abbreviation of MAPKK
Response: Thanks for the reviewer’s suggestion. We have added the full name of MAPKK (Mitogen activated protein kinase kinase). (Now in lines 21-22).
- L 23, please write the abbreviation of stMAPK
Response: Thanks for the reviewer’s suggestion. We have added the full name of Solanum tuberosum MAPK (StMAPK). (Now in lines 24).
- In the whole article, the genes names and botanical names of crops/plants must be italic, and protein names must be non-italic
Response: Many thanks for the reviewer’s constructive suggestion. We have checked the whole text carefully to make sure that all gene names are italics text, and remain plain text only when it was about proteins.
- L 97, Quantitative real-time PCR (qRT-PCR)
Response: Many thanks for the reviewer’s constructive suggestion. We have added Quantitative real-time PCR. (Now in line 100).
- Material and Methods; the author must write the accessed date of online websites or software, e.g. 05 December 2022, when the author used the online website.
Response: Thanks for the reviewer’s suggestion. Access dates that we have added.
Accordingly, the sentence in lines 106-109 “The potato genome sequences (http://spuddb.uga.edu/) was used to identify MAPK and MAPKK genes. Hidden Markov model search software (HMM) was used to identified the conserved domain of MAPK protein (http://pfam.xfam.org/family/PF00069), the E-value was set to 0.01 [20-21].” was revised as “The conserved domain of MAPK proteins (http://pfam.xfam.org/family/PF00069; accessed on 13 October 2021) was used as the probe to search StMAPK and StMAPKK sequnences in the potato Genome Database (http://spuddb.uga.edu/; accessed on 5 April 2021) using the Hidden Markov model search software (HMM) with the E-value set to 0.01 [20-21].” (Now in lines 108-112).
the Paragraph 2.3 was revised as “The intron and exon structure of the StMAPK and StMAPKK gene were obtained by comparing the coding sequence with the corresponding genomic DNA sequence by using Gene Structure Display Server (http://gsds.cbi.pku.edu.cn/; accessed on 3 December 2021) [23].The MEME software (http://meme.sdsc.edu/meme/cgi-bin/meme.cgi; accessed on 3 December 2021) was used to analyze StMAPK and StMAPKK genes conserved motifs, where the maximum motif number was set to 10, the optimal the motif width was set to 6-50 [24]. In addition, PlantCARE (http://bioinformatics.psb.ugent.be/webtools/plantcare/html/; accessed on 6 December 2021) database was used to predict cis acting elements of MAPK and MAPKK genes promoter region in potato.” (Now in lines 128-137).
the Paragraph 2.5 was revised as “The ProtParam program on the online software ExPASy website (http://web.expasy.org/protparam/; accessed on 15 December 2021) was used to predict StMAPK and StMAPKK protein signatures, including the number of amino acids (AA), theoretical molecular weight (MW), instability index (II), and isoelectric point (pI).” (Now in lines 149-152).
We have added access dates for online websites or software
Accordingly,the line 112 “(https://www.ncbi.nlm.nih.gov/)” was revised as “(https://www.ncbi.nlm.nih.gov/; accessed on 25 November 2021)”(Now in line 114).
the line 113 “(http://www.ebi.ac.uk/interpro/)” was revised as “(http://www.ebi.ac.uk/interpro/; accessed on 25 November 2021)”(Now in line 115).
the line 114 “(http://www.arabidopsis.org/)” was revised as “(http://www.arabidopsis.org/; accessed on 15 March 2022)”(Now in line 117).
the line 118 “(https://pir.georgetown.edu/pirwww/search/multialn.shtml)” was revised as “(https://pir.georgetown.edu/pirwww/search/multialn.shtml; accessed on 3 December 2021)”(Now in lines 121-122).
the line 132 “(http://pdgd.njau.edu.cn:8080/)” was revised as “(http://pdgd.njau.edu.cn:8080/; accessed on 20 March 2022)”(Now in line 140).
- L 145 needs space between 16h, 24h, etc.; please check the whole MS
Response: Thanks for the reviewer’s suggestion. We have added spaces in the middle of 16, 24h (Now in line 156), and checked the entire manuscript for revision.
- L 156-157, no need to write about the company; just write about software, which software or how to design the primers for expression analysis.
Response: Thanks for the reviewer’s suggestion. the sentence in lines 156-157 “qRT-PCR primers were synthesized by Tsingke Biotechnology Co., Ltd (Table S1).” was revised as “Specific primers for StMAPK and StMAPKK genes were designed using Beacon Designer 7.7 (Premier Biosoft International, Palo Alto, CA, USA) (Table S1).” (Now in lines 169-171).
- L 164, please delete one dot.
Response: Thanks for the reviewer’s suggestion. We have deleted the dot at the end of the line 178.
- Please carefully read the MS and check the space, dot, etc
Response: Many thanks for the reviewer’s constructive suggestion. We are sorry for these specifics errors. In this revised manuscript, we corrected these errors and checked space and dot throughout this paper.

Round 2
Reviewer 1 Report
The authors answered my questions comprehensively. The corrections made in the manuscript are fully sufficient.